# *VFL-Cafe*: Communication-Efficient Vertical Federated Learning via Dynamic Caching and Feature Selection

**DOI:** 10.3390/e27010066

**Published:** 2025-01-14

**Authors:** Jiahui Zhou, Han Liang, Tian Wu, Xiaoxi Zhang, Yu Jiang, Chee Wei Tan

**Affiliations:** 1School of Computer and Science and Engineering, Sun Yat-sen University, Guangzhou 510275, China; zhoujh77@mail2.sysu.edu.cn (J.Z.); liangh68@mail2.sysu.edu.cn (H.L.); wutian@mail2.sysu.edu.cn (T.W.); 2College of Computing and Data Science, Nanyang Technological University in Singapore, Singapore 639798, Singapore; yu012@e.ntu.edu.sg (Y.J.); cheewei.tan@ntu.edu.sg (C.W.T.)

**Keywords:** vertical federated learning, communication efficient, feature selection, dynamic caching

## Abstract

Vertical Federated Learning (VFL) is a promising category of Federated Learning that enables collaborative model training among distributed parties with data privacy protection. Due to its unique training architecture, a key challenge of VFL is high communication cost due to transmitting intermediate results between the Active Party and Passive Parties. Current communication-efficient VFL methods rely on using stale results without meticulous selection, which can impair model accuracy, particularly in noisy data environments. To address these limitations, this work proposes *VFL-Cafe*, a new VFL training method that leverages dynamic caching and feature selection to boost communication efficiency and model accuracy. In each communication round, the employed caching scheme allows multiple batches of intermediate results to be cached and strategically reused by different parties, reducing the communication overhead while maintaining model accuracy. Additionally, to eliminate the negative impact of noisy features that may undermine the effectiveness of using stale results to reduce communication rounds and incur significant model degradation, a feature selection strategy is integrated into each round of local updates. Theoretical analysis is then conducted to provide guidance on cache configuration, optimizing performance. Finally, extensive experimental results validate *VFL-Cafe*’s efficacy, demonstrating remarkable improvements in communication efficiency and model accuracy.

## 1. Introduction

Vertical Federated Learning (VFL) [1,2,3,4] is an important category of the Federated Learning (FL) paradigm [5,6,7,8] that promotes joint training of models among multiple distributed participants while protecting data privacy. Unlike a more conventional category, usually termed as Horizontal Federated Learning (HFL), VFL is performed by parties who hold disparate and complementary features but share the same training sample ID space. In contrast, HFL parties possess different sample IDs in their local datasets sharing the same feature space. Another key difference is that VFL’s training participants consist of a single *Active Party,* which is the only party possessing the labels of the training data, while all the others are *Passive Parties*. Each of the HFL parties, instead, owns its local datasets containing both features and labels. This characteristic also leads to a unique model architecture and interaction patten among parties, differing from that in HFL frameworks.

As shown in Figure 1, the Active Party, which is the only party that owns the labels, deploys a top model, which usually consists of the final layer(s) of the complete model, and a bottom model, which comprises the initial layers of the complete model. Differently, each Passive Party owns only its local bottom model. Given this architecture, in each forward pass, activations are passed through each bottom model concurrently and independently, and the activations of the last layer of each bottom model, termed as *embeddings*, are required to be transmitted to the Active Party and will pass through the top model, producing the final model output, i.e., predictions. In each backward pass, the *gradients of the top model* calculated by the Active Party, based on the losses from predictions and corresponding labels, are transmitted back to each Passive Party, to enable continued gradient computations in the bottom models. These transmitted embeddings and gradients of the top model are referred to as *intermediate results* in the rest of this paper. In contrast, the transmitted results among HFL participants are the parameters of each participant’s complete model, which are independently trained at each participant, before each communication round, for synchronizing the local models.

Challenges of VFL. Apparently, despite its advantage of privacy preservation among different parties, VFL faces a primary challenge of incurring extensive communications between the Active and Passive Parties, i.e., transmitting intermediate results in both forward and backward passes. In addition, different parties in practice tend to be geo-distributed with heterogeneous network infrastructures and typically communicate through WAN connections [9,10]. Therefore, frequent communications yield expensive communication latency and cost. In addition, another challenge usually overlooked by existing VFL studies is that the presence of noisy features unrelated to the prediction task can further escalate communication costs and local computation demands, ultimately reducing prediction accuracy. Therefore, it is imperative to develop communication-efficient methods for VFL training while achieving acceptable model accuracies.

Communication Mitigation. A large number of FL studies have been devoted to reducing the number of communication rounds through additional local updates using stale results received from other parties [11,12,13]. However, most of these works focus on HFL settings, where both the labels and features of individual data samples are stored locally, allowing each participant to independently perform forward and backward propagations. In contrast, the stale embeddings and gradients of VFL training processes are separately obtained and transmitted by different parties, which makes the implementation of communication mitigation schemes more complex. Recently, a few studies proposed to use stale intermediate results for local updates in VFL training. For instance, Liu et al. [14] propose FedBCD, a well-known VFL method that enables each party to reuse the intermediate results of each batch multiple times and leverages estimated gradients to update the model, enabling multiple local updates before each communication to reduce the transmission overhead. FlexVFL [15] and AdaVFL [16] further analyze the relationship between model convergence and the number of local updates, assigning various numbers of local updates to different Passive Parties based on their computation and communication capacities.

However, these pioneering communication-efficient VFL methods use stale and approximated intermediate results, which may compromise the final model accuracy. Moreover, this performance degradation can be more severe when local datasets include noisy features. In order to overcome the above problems, we propose a new communication-efficient VFL method, named *VFL-Cafe*, based on dynamic caching and feature selection. It can effectively reduce communication overhead and boost model accuracy, particularly in scenarios involving noisy data features and large communication latencies (e.g., due to high transmission delay and/or propagation delay in the WAN connections).

The main contributions of this work are summarized as follows:We propose a new VFL training framework, *VFL-Cafe*, to effectively reduce the number of communication rounds by transmitting multiple batches of intermediate results in each communication and strategically using stale results. The first technique employed is a dynamic caching scheme. Given this, the transmitted results are cached locally with dynamic replacement, allowing each party to selectively use the stale results in the cache. In addition to reducing the communication overhead, this method can achieve the expected training accuracy with a limited number of communications.An in-depth theoretical analysis is conducted for configuring the cache system, which is related to choosing different parameters to set up the cache and the number of batches transmitted in one communication. This analysis provides guiding principles for the design and parameter tuning of the cache system, which can also help clarify its impact on communication efficiency and model performance.We also employ a feature selection strategy for sparse weights of the underlying model and integrate it with the caching scheme within each communication round. By reducing the impact of noisy features on the model, our method effectively improves the robustness and prediction accuracy of the model. This is particularly helpful for scenarios with redundant and noisy features in the datasets, where using staled intermediate results can severely impede model accuracy.Extensive experimental results conducted on our VFL testbed demonstrate the efficiency and effectiveness of our algorithm *VFL-Cafe*. We achieve up to 46.87% communication rounds reduction while improving the test accuracy by up to 79.6%.

The remainder of this paper is structured as follows. Section 2 provides an overview of related research in this area. Section 3 introduces our proposed approach, which is to sparsify the weights of model parameters by transferring multiple intermediate results at a time through a cache system. Section 4 explores the relationship between the parameters of the cache system and the number of intermediate results transferred at a time. Finally, Section 5 draws conclusions from our findings and provides suggestions for future research directions.

## 2. Related Work

In this section, we first introduce the related works on VFL mechanisms with multiple local updates per communication round and then provide references on feature selection schemes applied to VFL training frameworks.

Different from Horizontal Federated Learning [7,13] , which reduces communication costs by aggregating models trained on distributed datasets with a shared feature space across participants [11,17,18,19,20,21], Vertical Federated Learning faces distinct challenges due to its vertically partitioned data structure. In VFL, participants collaborate on a shared set of samples with unique and non-overlapping features, which limits the direct application of HFL strategies. As a result, VFL requires tailored approaches to address the communication overhead associated with processing and transferring heterogeneous feature representations [1,2,3,4].

In this section, we first provide an overview of VFL methods focusing on data compression and sample selection techniques developed to mitigate communication overhead. Following this, we discuss multiple local update strategies and specialized feature selection mechanisms that have been integrated into VFL frameworks to further improve computational and communication efficiency.

### 2.1. Vertical Federated Learning

VFL has garnered significant attention due to its applicability in scenarios where feature distribution is vertically partitioned across parties. To address communication overhead, data compression and quantization have been widely adopted. AVFL [22] uses principal component analysis (PCA) for dimensionality reduction, while CE-VFL [23] combines PCA with autoencoders to learn latent representations, reducing data transmission. SecureBoost [24] encodes encrypted gradients into single messages, minimizing encryption costs and transmission volume. GP-AVFL [25] employs bidirectional sparse compression techniques to reduce communication data. Furthermore, T-VFL [26] adopts a truncation strategy, discarding parties with low channel gain to lower training latency.

Several studies focus on client or sample selection to improve VFL efficiency. For instance, Jiang J. et al. [27] leverage mutual information for participant selection, optimizing party-level evaluation to enhance VFL scalability. Yang et al. [28] use model compression to transmit only significant updates above a threshold, with residual updates locally accumulated. These studies collectively underscore efforts in data compression, quantization, and sample selection, with an emphasis on communication efficiency.

### 2.2. Multiple Local Updates

Multiple local updates are a common strategy in VFL to reduce communication overhead by allowing clients to perform several local gradient updates before communication. Liu et al. [29] propose the FedBCD algorithm, which reduces communication overhead effectively but encounters issues with gradient variance and delayed convergence due to stale data. To address these, adaptive methods like Flex-VFL [15] adjust the learning rate according to operational speed and parameters, while AdaVFL [16] selects local update frequencies adaptively. Xie et al. [30] introduce VIM, an ADMM-based optimization framework with differential privacy that enhances performance and reduces communication through multiple local updates. CELU-VFL [10] improves FedBCD with caching and polling strategies to mitigate stale result accumulation while maintaining efficiency. Despite these advancements, challenges remain in handling noisy datasets and fluctuating network environments.

### 2.3. Feature Selection

Feature selection aims to reduce redundant features and lower data volume, thus saving communication costs. FedSDG-FS [31] employs Gaussian random double gates and Gini impurity-based feature initialization to reduce computational load. LESS-VFL [32], after a short pre-training cycle, optimizes part of its global model to determine the relevant outputs of each party’s model. This information is shared with all parties, allowing local feature selection without the need for the parties to communicate. Feng S. et al. [33] utilizes non-overlapping data and integrates autoencoders and L2 norms to embed importance representations for feature selection. By focusing on important features, these methods achieve efficient training with minimal data exchange.

### 2.4. Summary of Differences Compared with This Work

Although many VFL methods employ multiple local updates with stale gradients or intermediate results, most do not selectively cache these results. Additionally, the potential of integrating feature selection with caching is largely unexplored. This study addresses these gaps by introducing a method that significantly enhances communication efficiency, setting it apart from existing techniques.

## 3. Primer on VFL: Problem Setting and Formulation

Vertical Federated Learning (VFL) enables collaborative model training across multiple parties while preserving data privacy. In a typical scenario, we consider *K* data parties, where each party holds a unique subset of features from *N* samples, denoted as ${\left\{x_{i},y_{i}\right\}}_{i=1}^{N}$. The feature vector $x_{i}\in R^{1\times d}$ is partitioned among the parties such that party *k* possesses features $x_{i}^{k}\in R^{1\times d_{k}}$, with $d_{k}$ representing the feature dimension. We assume that the label information is exclusively held by the last party, *K*. The collective dataset can be represented as follows:$$D_{k}={\left\{x_{i}^{k}\right\}}_{i=1}^{N},\;D_{K}={\left\{x_{i}^{K},y_{i}\right\}}_{i=1}^{N},\;D={\left\{D_{k}\right\}}_{k=1}^{K}.$$

The training process involves both the Active Party and Passive Parties. The Active Party possesses a local model (bottom model) as well as a top model, which aggregates the outputs from the Passive Parties’ bottom models. Thus, each party has the capability to perform local computations on its data while also contributing to the overall model via the top model.

The challenge lies in enabling each party to compute their model parameters $\theta_{{\mathrm{Bot}}_{k}}$ without sharing their local datasets $D_{k}$ or model parameters with others. To formulate the collaborative training process, we aim to minimize the overall loss across all parties, represented mathematically as:(1)$$\min_{\Theta }L\left(\Theta ;D\right)=\frac{1}{N}\sum_{i=1}^{N}f\left(\theta_{{\mathrm{Bot}}_{1}},\ldots ,\theta_{{\mathrm{Bot}}_{K}},\theta_{\mathrm{Top}};D_{i}\right)+\lambda \sum_{k=1}^{K}\gamma \left(\left(\theta_{{\mathrm{Bot}}_{k}}\right)+\gamma \theta_{\mathrm{Top}}\right),$$
where $\Theta =[\theta_{{\mathrm{Bot}}_{1}};\ldots ;\theta_{{\mathrm{Bot}}_{K}}]$ comprises the parameters from all parties’ bottom models, $\theta_{\mathrm{Top}}$ denotes the parameters of Active Party’s top model, f(·) is the loss function measuring the prediction error, and γ(·) is a regularization term to prevent overfitting.

During the training phase, the parties exchange only the necessary intermediate results, specifically the local model outputs $H_{k}=G_{k}\left(D_{k}^{B},\theta_{k}\right)$ computed on mini-batches of samples $\left(D_{k}^{B}={\left\{x_{i}^{k}\right\}}_{i=1}^{B}\right)$ and the corresponding gradients $\nabla H_{k}$. This approach ensures the preservation of privacy throughout the training process. The Active Party receives the outputs from the Passive Parties, denoted as $H_{1},H_{2},\ldots ,H_{K-1}$, in addition to its own output $H_{K}$. These outputs are aggregated within the Active Party’s top model to compute the global loss and gradients, which then are transmitted back to the Passive Parties for local parameter updates:(2)$$\nabla \theta_{{\mathrm{Bot}}_{k}}L=\frac{\partial L}{\partial \theta_{{\mathrm{Bot}}_{k}}}=\frac{\partial L}{\partial H_{k}}\frac{\partial H_{k}}{\partial \theta_{{\mathrm{Bot}}_{k}}}.$$

This process is iterated until convergence. In summary, the VFL framework allows for collaborative learning while maintaining data privacy. The Active Party’s dual role as both a contributor and an aggregator of information is essential for effective model training in environments with distributed data.

## 4. Our Algorithm *VFL-Cafe*: A Nice Integration of Dynamic Caching and Feature Selection

In the preceding sections, we focus on communication inefficiencies and the challenges posed by noisy datasets, aiming to overcome the limitations of conventional VFL approaches. In a nutshell, our algorithm *VFL-Cafe* enables parallel execution of communication and strategical local updates, which nicely integrates caching, intermediate result sampling, and feature selection.

By examining existing strategies for multiple local updates and feature selection, we overcome the limitation of traditional VFL, where only one mini-batch is transmitted per communication and stale intermediate results are not strategically used. Our method, as shown in Figure 2, exchanges the intermediate statistics of multiple mini-batches within a single communication. These statistics are stored in a cache work table with the support of the cache system to obtain fresher intermediate results. During local updates, we adopt the caching mechanism [10] where the intermediate results obtained from each communication are dynamically stored in the cache work table. A Round-Robin strategy is also applied for sampling the intermediate results. At the same time, the staleness of cached results is measured by calculating the cosine similarity of forward activations (or backward gradients), enabling asynchronous local updates with weighted adjustments. Moreover, the local updates method at each party is based on the sampled results from the cache and feature selection procedures.

More formally, in our framework, a Passive Party retains the bottom model parameters $\theta_{{\mathrm{Bot}}_{P}}$, while the Active Party manages both its bottom model parameters $\theta_{{\mathrm{Bot}}_{A}}$ and a top model, along with access to the corresponding data labels. The core idea of the proposed Algorithm 1 is to treat *N* mini-batches as a batch group, leveraging both communication and local update threads to complete model updates efficiently.
**Algorithm 1** Communication: VFL training with grouped mini-batches updates  1:**for** each group of *N* mini-batches **do**  2:      **if** Is Passive Party **then**  3:           Compute and send $H_{P}$;  4:           Recv $\nabla H_{P}$;  5:           Update with $\nabla \theta_{{\mathrm{Bot}}_{P}}=\sum_{i=1}^{N}\nabla H_{P}^{\left(i\right)}\frac{\partial H_{P}^{\left(i\right)}}{\partial \theta_{{\mathrm{Bot}}_{P}}}$;  6:           Put $H_{P}$ and $\nabla H_{P}$ into Cache in order  7:      **end if**  8:      **if** Is Active Party **then**  9:           Compute $H_{A}$ and Recv $H_{P}$ for the group;10:           **for** each mini-batch *k* in the group **do**11:                Feed $H_{P}$ and $H_{A}$ into top model, perform backward propagation;12:                Update $\theta_{\mathrm{Top}}$ and $\theta_{{\mathrm{Bot}}_{A}}$;13:                Compute $\nabla H_{P}^{\left(i\right)}$;14:           **end for**15:           Send $\nabla H_{P}$ for the group;16:      **end if**17:      Put $H_{P}$ and $\nabla H_{P}$ into Cache in order18:**end for**

Communication Process. We start training with *N* mini-batches treated as a group, shown in Algorithm 1. Each Passive Party computes the forward activation output $H_{P}$ on $\theta_{{\mathrm{Bot}}_{P}}$ for *N* mini-batches and transmits $H_{P}$ to the Active Party in a single communication step (lines 2–3). For ease of representation, we omit the index (*k*) of the Passive Parties in this section, as the proposed algorithms are uniform across all Passive Parties. Simultaneously, the Active Party computes the forward activation output $H_{A}$ on $\theta_{{\mathrm{Bot}}_{A}}$ for the same *N* mini-batches. Upon receiving the intermediate activation $H_{P}$ from the Passive Party, the Active Party sequentially feeds these data into the top model for backward propagation. During this process, the Active Party updates both the top model parameters and $\theta_{{\mathrm{Bot}}_{A}}$ (lines 8–14). After completing the computations, the Active Party transmits the backward gradients $\nabla H_{P}$ (line 15) with respect to $\theta_{{\mathrm{Bot}}_{P}}$ for the *N* mini-batches back to the Passive Party, allowing it to update $\theta_{{\mathrm{Bot}}_{P}}$ (lines 4–5). Following one complete communication cycle, $H_{P}$ and $\nabla H_{P}$ are cached at the corresponding party locally for use in local updates.

Local Update Process. As shown in line 2 of Algorithm 2, the Active Party (and the Passive Party, if applicable) retrieves the stale forward activation ${\tilde{H}}_{P}$ and its corresponding backward gradient $\nabla {\tilde{H}}_{P}$ from the cache. It then computes the fresh forward activation $H_{P}$ and backward gradient $\nabla H_{P}$ for the associated mini-batch (line 7 and line 15). The stale activations and gradients, ${\tilde{H}}_{P}$ and $\nabla {\tilde{H}}_{P}$, are used to calculate the cosine similarity Ins_weights with the fresh ones, $H_{P}$ and $\nabla H_{P}$ (line 8 and line 16). For instance, considering the Active Party, the function of calculating the cosine similarity between two vectors $H_{P}$ and ${\tilde{H}}_{P}$ is in a batch-wise manner:(3)$$\cos\text{\_}\mathrm{similarity}\left(H_{P},{\tilde{H}}_{P}\right)=\frac{\sum H_{P,i}{\tilde{H}}_{P,i}}{\max\left(\sqrt{\sum H_{P,i}^{2}}\cdot \sqrt{\sum {\tilde{H}}_{P,i}^{2}},\;\epsilon \right)}.$$

Here, $\sum H_{P,i}{\tilde{H}}_{P,i}$ represents the dot product of vectors $H_{P}$ and ${\tilde{H}}_{P}$, while $\sqrt{\sum H_{P,i}^{2}}$ and $\sqrt{\sum {\tilde{H}}_{P,i}^{2}}$ denote the L2 norms (magnitudes) of $H_{P}$ and ${\tilde{H}}_{P}$, respectively. A small constant ϵ is added to avoid division by zero. To ensure the result remains within the valid range of cosine similarity, the value is clipped between −1.0 and 1.0.
**Algorithm 2** LocalUpdates: Using cached intermediate results and feature selection  1:**while** training **do**  2:      Extract stale ${\tilde{H}}_{P}$ and $\nabla {\tilde{H}}_{P}$ from Cache of mini-batch *i*.  3:      **if** Is Passive Party **then**  4:            **if** communication round reaches threshold **then**  5:                  Apply sparsification on the first layer of $\theta_{{\mathrm{Bot}}_{P}}$ to select features.  6:            **end if**  7:            Compute current $H_{P}$;  8:            Calculate cosine similarity Ins_weights between ${\tilde{H}}_{P}$ and $H_{P}$.  9:            Update with Ins_weights⊙${\tilde{H}}_{P}$.10:      **end if**11:      **if** Is Active Party **then**12:            **if** communication round reaches threshold **then**13:                  Apply sparsification on the first layer of $\theta_{{\mathrm{Bot}}_{A}}$ to select features.14:            **end if**15:            Compute current $\nabla H_{P}$;16:            Calculate cosine similarity Ins_weights between $\nabla {\tilde{H}}_{P}$ and $\nabla H_{P}$.17:            Update with Ins_weights⊙${\tilde{H}}_{P}$.18:      **end if**19:**end while**

This similarity is employed to adjust the weight of the cached instances according to their staleness, preventing instances from becoming overly outdated. After several communication rounds (line 9 and line 17), the first layer of the bottom model parameters $\theta_{{\mathrm{Bot}}_{P}}$ and $\theta_{{\mathrm{Bot}}_{A}}$ undergoes sparsification to enhance feature selection and model efficiency.

Feature Selection. In our feature selection mechanism, we focus on the first-layer network parameters of the bottom models for both the Passive Parties and Active Party. This process involves the sparsification of weights after a predefined number of communication rounds, indicating that the network has completed its pre-training phase, as shown in line 4–6 of Algorithm 2. Specifically, we compute the L2 norm of first-layer network parameters corresponding to the each feature, denoted as $∥w_{j}{∥}_{2}$ for each feature *j*. The sparsification process is defined by comparing the L2 norm of each $∥w_{j}{∥}_{2}$ to a specified threshold τ:(4)$$\mathrm{If}\;{∥w_{j}∥}_{2}<\tau ,\;\mathrm{then}\;\mathrm{set}\;w_{j}=0.$$

This criterion effectively filters out features that contribute minimally to the model, thus facilitating the selection of the most relevant features. By implementing this feature selection mechanism, we not only reduce the number of active parameters but also significantly lower the computational burden during both training and inference phases. Consequently, this approach enhances the efficiency of the model while maintaining performance, contributing to the overall effectiveness of our VFL framework.

## 5. Analysis for Configuring the Caches and the Number of Batches per Communication

Note that our communication-efficient strategy based on caching also requires proper configuration of the local caches and the dynamic process of retrieving and using each of the batches stored in the caches. Intuitively, a larger number of times that each batch of stale intermediate results can be used for local updates (*R*) yields a larger staleness and thus more severe potential model degradation; but a sufficiently large *R* is also required since there might be an insufficient number of batches ready in the cache, which depends on the computation speed, the number of batches per communication *N*, and the communication latency. In addition, the total number of updates per round (*Q*) has a similar effect on model accuracy, as more updates may incur higher staleness; but a larger *Q* can reduce the communication frequency, therefore boosting the communication efficiency. Differently, the number of mini-batches *N* as a group that each party transmit per round is leveraged to enable pipeline parallelism that can overlap computation and communication, while making full use of the transmission bandwidth. Note that although sending *N* batches in a group per time yields a similar transmission data volume to sending *N* batches sequentially with each batch transmitted once the corresponding backward or forward pass using the previous batch finishes, the total latency is largely reduced by batching. Correspondingly, the total length of the cache, *W*, should be large enough for the required local updates and number of batches that need to be stored in each communication.

Given the above complex effects of different parameters and their relationships, we take the first attempt to formalize our understanding of the cache configuration in this section. This theoretical analysis is meant to guide the implementation of *VFL-Cafe* and help the readers to understand the caching with batching strategy. Given the number of mini-batches *N*(≥2) transmitted in a single communication, we can analyze the relationship between the number of consecutive local updates *Q* per round (by using all the available batches of intermediate results in the cache), the length of the cache working set *W*, the maximum number of times a certain mini-batch can be sampled from the working set *R*, and the number of mini-batches *N* transmitted in each communication. This section aims to explore optimization strategies for communication intervals and caching in computational networks. We first list our basic observations as follows.

To achieve Round-Robin local sampling, when transmitting *N* mini-batches in a single communication, subsequent mini-batches must satisfy the condition of completing *R* updates before being removed from the cache working set.Initially, if the cache working set is not yet full, only the updates from the current communication mini-batches will be performed.Each mini-batch in the cache cannot be sampled again in the next W−1 steps.

In terms of implementation, given *N* and *R*, when setting Q=N×R, we can derive the minimum value of *W* that satisfies this condition. We will discuss three different scenarios and provide programming analysis for the latter two cases.

### 5.1. Case 1: W ≥ Q

According to the Round-Robin local sampling strategy, we select *Q* mini-batches from the window *W* for updating. Since Q≤W, it suffices to satisfy Q=N×R to ensure that subsequent mini-batches can undergo *R* updates before being removed from the cache. This condition guarantees that *W* remains in a stable equilibrium, allowing for *Q* updates per communication round. We can view both *W* and *Q* as fixed windows (refer to the blue box labeled with “Cache” and the brown box labeled with “Local Updates” in Figure 3, where *W* and *Q* are their heights, respectively). We refer to *W* as the main window and Q−W as the virtual window. Mini-batches are treated as flowing data; only those existing within the main window have the opportunity for updates. The number of new mini-batches added each time is *N*, and in order to utilize *R* updates effectively, we must have Q=N×R. This condition ensures that the total number of pending updates in the *W* window remains constant, as the newly added *N* mini-batches will collectively increase the cache working set by N×R updates, perfectly counterbalancing the *Q* updates consumed in this round.

As illustrated in Figure 3, we set the cache with parameters W=8, R=3, Q=6, and N=2, matching the conditions W≥Q and Q=N×R. Both the fifth and sixth mini-batches can precisely complete *R* updates in the cache working set. Similarly, the seventh and eighth mini-batches also meet the requirement for *R* updates, thereby ensuring that subsequent mini-batches can also fulfill the exact completion of *R* updates, enabling the Round-Robin local sampling strategy.

### 5.2. Case 2: Q > W ≥ 0.5 × Q

We utilize the following formula to assess whether the corresponding mini-batches (the left-hand side of inequality (Equation 5)) can achieve *R* updates:(5)$$\left[\frac{\left(W-i+d_{1}\right)}{N}\right]+1+\left[\frac{\left(Q-W-i+d_{2}\right)}{N}\right]+1\geq R,$$
where $d_{1}$ and $d_{2}$ represent the number of mini-batches that completed updates ahead of time, starting at 0. The components $\left[\frac{\left(W-i+d_{1}\right)}{N}\right]+1$ and $\left[\frac{\left(Q-W-i+d_{2}\right)}{N}\right]+1$ denote the number of updates that the *i*-th mini-batch can complete in the main and virtual windows, respectively. If the sum of these quantities is greater than or equal to *R*, it indicates that the mini-batch can satisfy the completion of *Q* updates. The condition holds only if all *N* mini-batches can satisfy the Formula (Equation 5). We can validate this by writing a program to automatically determine the minimum value of *W* satisfying the conditions for a given *N* and *R*, with Q=N×R.

Our implemented program has the following basic steps as follows:Number the subsequent *N* mini-batches from 1 to *N*, starting from the smallest index;Use Formula (Equation 5) to verify if *R* updates can be met; if so, identify the round *t* in which *R* updates are completed;For the subsequent mini-batches in round *t*, check their window placement:
-If they are also in the virtual window, increment $d_{1}$ and $d_{2}$;-If not in the virtual window, only increment $d_{1}$;Repeat this process until all mini-batches conform to Formula (Equation 5); any deviations indicate an invalid *W* value.

As shown in Figure 4, when W=5, R=4, Q=8, and N=2, the condition Q>W≥0.5×Q is satisfied, indicating that the virtual window exceeds the main window. In this scenario, when Q=N×R, both the fifth and sixth mini-batches can complete *R* updates in the cache working set. However, the fifth mini-batch’s opportunity arises from the sixth mini-batch having completed *R* updates after the previous fourth round of communication, resulting in its removal from the cache working set. Consequently, the fifth mini-batch can complete its *R* updates after the fifth round of communication. Similarly, the seventh and eighth mini-batches also satisfy the *R* updates, ensuring that this parameter configuration enables the Round-Robin local sampling strategy.

### 5.3. Case 3: 0.5 × Q > W ≥ 2

In this case, the virtual window is larger than the main window. If no mini-batches complete *Q* updates in this round, the virtual window comprises both the completely repeated sections of the main window and potentially incomplete repetitions. First, we calculate the occurrence of the main window $n=\left[\frac{\left(W-i+d\right)}{N}\right]+1$.

We utilize the following formula for assessment:(6)$$\sum_{k=1}^{n}\left(1+\left\lfloor \frac{\left(Q_{i-1}-W\right)}{W}\right\rfloor \right)+\left\lfloor \frac{(Q_{i-1}-W)\;\mathrm{mod}\;W}{\left(i+(t-1)\times N-d\right)}\right\rfloor \geq R,$$
where *n* indicates the number of times the mini-batch can appear in the main window, denoting the maximum rounds it can undergo. $Q_{k}$ signifies the position when the i−1-th mini-batch completes *R* updates, defaulting to *Q*. The first term indicates the appearance of the mini-batch in the main window during this round, while the second term reflects the number of complete repetitions of the main window in the virtual window. The third term represents the incomplete sections of the main window in the virtual window. Here, *t* represents the current round, which intuitively signifies whether the actual position (denominator) exceeds the length of the incomplete section; if it does not, an update is permissible. We can validate this through programs to automatically decide the minimum *W* value that satisfies the condition for a given *N* and *R* with Q=N×R.

Our implemented program has the following basic steps as follows:Number the subsequent *N* mini-batches from 1 to *N*, starting from the smallest index;Use Formula (Equation 6) to check if *R* updates can be met; if so, record the round *t* and position $Q_{i}$ where *R* updates are completed;For the subsequent mini-batches, if in round *t* they are in the main window, increment *d*;Continue until all mini-batches satisfy Formula (Equation 6); if any do not, the value of *W* is invalid.

In summary, through theoretical analysis, we can reasonably select parameters to maintain communication efficiency while avoiding repeated use of outdated mini-batch data, thus enhancing both communication efficiency and model training speed.

## 6. Experimental Validation

In this section, we first introduce our experimental setup and then evaluate the performance of *VFL-Cafe* by comparing it with state-of-the-art methods and analyzing its robustness under various algorithmic parameters. We also present comparative results for *VFL-Cafe* under added noise conditions and assess the impact of feature selection by comparing AUC metrics before and after applying feature selection to *VFL-Cafe*.

### 6.1. Experiment Setup

Implementation: All experiments were conducted on two servers, referred to as Passive Party and Active Party. Each participant utilized an NVIDIA GeForce RTX 3090 GPU for training tasks. The proposed framework was implemented using TensorFlow and gRPC, with a lightweight message queue supporting inter-party communication. The FLGraph abstraction automates backward computation graph generation. Key features include customizable configuration files, user-friendly APIs (e.g., define_bottom_model()), and core services such as distributed training and secure communication.

Model: We employed the Deep Learning Recommendation Model, specifically the Wide and Deep Learning (WDL) architecture, which is commonly used in Vertical Federated Learning collaborations. More precisely, the guest-side bottom model consists of three fully connected layers with ReLU activation functions, sequentially transforming the input dimensions as [dim→256→256→256]. The host-side bottom model includes an embedding layer that maps categorical inputs into dense representations of size 128, followed by two fully connected layers transforming [dim×128→256→256]. The top model concatenates the outputs from the guest and host bottom models and passes the result through a single fully connected layer to produce the final prediction. The training process utilizes binary cross-entropy loss, which is standard in recommendation tasks such as click-through rate prediction:(7)$$L=-\frac{1}{N}\sum_{i=1}^{N}\left[y_{i}\log\left({\hat{y}}_{i}\right)+\left(1-y_{i}\right)\log\left(1-{\hat{y}}_{i}\right)\right],$$
where $y_{i}$ is the true label and ${\hat{y}}_{i}$ is the predicted probability obtained through the sigmoid function. This loss ensures effective optimization and robust evaluation of model performance.

Dataset: The experiments utilized the large-scale public Criteo dataset for advertising click-through rate prediction, which contains extensive information on online ad impressions and user clicks. This dataset, widely recognized as a benchmark in VFL research due to its large scale and diverse feature types, is particularly suitable for evaluating the proposed framework. It comprises 26 categorical features and 13 numerical features, with labels indicating whether an ad was clicked (1) or not clicked (0). The training and testing sets consist of 41 million and 4.5 million instances, respectively, with Participants A and B partitioning the data domains as 26 and 13 features.

Protocol: Model training was performed using the AdaGrad optimizer, with a mini-batch size of 4096 for experiments without noise. For experiments with added noise and those incorporating feature selection under noisy conditions, a mini-batch size of 512 was used. The output dimension of the forward activation $H_{P}$ was set to 256. Experimental data were averaged from three independent trials to mitigate potential errors.

### 6.2. End-to-End Evaluation

We compare our proposed algorithm, *VFL-Cafe*, against other baseline methods, including FedBCD with different local update counts (*R* = 5 and *R* = 8), CELU-VFL, and FedSGD. In the FedBCD method, R denotes the number of local updates conducted between each communication round. As shown in Figure 5a,b, when *R* = 8, FedBCD exhibits noticeable oscillations in the early stages, and its later performance is inferior to other models. This instability can be attributed to the increased gradient staleness from multiple local updates, which impacts the convergence stability of the model.

FedSGD, on the other hand, demonstrates relatively stable performance but converges significantly slower compared to CELU-VFL and our *VFL-Cafe*. Although it avoids the oscillations seen in FedBCD, its lower convergence speed limits its effectiveness.

In contrast, *VFL-Cafe* achieves faster convergence while maintaining higher AUC values over both communication rounds and running time. Compared to CELU-VFL and other baselines, *VFL-Cafe* provides a balanced approach with both stability and superior performance, underscoring its effectiveness in achieving efficient communication and robust model accuracy in VFL settings.

### 6.3. Ablation Experiment

Compared to traditional Vertical Federated Learning methods, the approach utilized in *VFL-Cafe* incorporates four additional techniques: transmission of multiple mini-batches, local updates, Round-Robin local sampling, and parameterized by *N*, *R*, and *W*. We will explore the impact of each technique on the results and evaluate their sensitivity to hyperparameters. Specifically, we trained the WDL model on the Criteo dataset and measured the number of communication rounds required to achieve the same AUC metric (set at 0.7950), as presented in Table 1.

Firstly, we assessed the effect of varying the number of mini-batches transmitted (N). Notably, when N=2, the number of communication rounds decreased by 6.64%, maintaining a similar AUC trend compared to single mini-batch transmission. As *N* increased to 3 and 4, the system demonstrated significant improvements in convergence speed, with communication rounds decreasing by 45.80% and 46.87%, respectively. This enhancement can be attributed to our configuration of W=8 and R=5, which allowed for the retrieval of more fresh intermediate results per communication. However, beyond a certain threshold of *N*, the limited cache size may lead to the premature removal of intermediate results that could benefit from further local updates, corroborating our earlier analysis regarding cache configuration and the number of mini-batches per communication.

Next, we examined the maximum number of times mini-batches can be sampled for local updates (*R*). Using R=1 as a baseline, we observed a substantial reduction of 58.97% in communication rounds when *R* was increased to 3, which further improved to approximately 59.64% at R=5. Although R=8 provided faster convergence in the early stages compared to R=5, the increased data staleness associated with larger *R* values resulted in similar communication round requirements to achieve the same AUC.

Lastly, we evaluated the impact of the Round-Robin local sampling strategy by varying *W*. With W=1 serving as a baseline, where only one mini-batch is retained in the cache, we found that this strategy consistently outperformed sequential sampling. The required communication rounds decreased by approximately 40% across various *W* values. Additionally, for *W* values of 3, 5, and 8, the overall convergence speed remained relatively stable, indicating that the choice of cache working set size exhibits robustness in our approach.

### 6.4. Denoising Experiment

In the noisy setting, we introduced Gaussian noise equivalent to the original data volume, with a mean and variance matching those of the original dataset. Gaussian noise was chosen due to its widespread use in VFL feature selection and its statistical properties, which make it a standard choice for simulating realistic noise conditions. This noise configuration allowed us to rigorously evaluate the robustness of each method under challenging conditions.

Under noisy conditions, *VFL-Cafe* demonstrates a clear performance advantage over other baseline methods shown in Figure 6a. Compared to FedSGD, FedBCD (*R* = 5), and FedBCD (*R* = 8), the AUC curve of *VFL-Cafe* not only rises to a high level rapidly in the early stages but also maintains stability and low fluctuation throughout the training process. Although FedBCD (*R* = 8) and CELU show some upward trend initially, their final AUC remains low with considerable oscillations, indicating instability under noise. The FedSGD methods still fail to effectively mitigate the impact of noise, resulting in suboptimal convergence. Our method, *VFL-Cafe*, with its faster convergence and higher final AUC, demonstrates superior robustness and stability under noisy conditions.

As shown in Figure 6b, the addition of feature selection in *VFL-Cafe* further enhances performance by filtering out redundant features. This feature selection mechanism effectively removes 100% of redundant features of the Passive Party, thereby significantly improving the model’s noise resilience and overall stability. The resulting AUC curve confirms the efficacy of feature selection, as *VFL-Cafe* achieves a faster convergence rate and sustains a higher AUC compared to the configuration without feature selection.

## 7. Conclusions

This study introduces *VFL-Cafe*, a novel approach to Vertical Federated Learning that addresses challenges in communication efficiency and feature selection. By leveraging dynamic caching and a strategic feature selection mechanism, *VFL-Cafe* reduces communication overhead and improves model accuracy in both noisy and noise-free environments. Experimental results highlight that *VFL-Cafe* achieves substantial reductions in communication latency while maintaining predictive performance, demonstrating its effectiveness over traditional VFL methods.

*VFL-Cafe* reduces communication overhead and improves model accuracy in both noisy and noise-free environments. Future directions could focus on refining feature selection to enhance model robustness in diverse data environments and integrating advanced privacy-preserving techniques, such as differential privacy and secure multi-party computation, to further protect data confidentiality. Additionally, efforts toward model explainability and fairness will be crucial, ensuring transparent and equitable decision making across diverse data distributions.

## Figures and Tables

**Figure 1 entropy-27-00066-f001:**
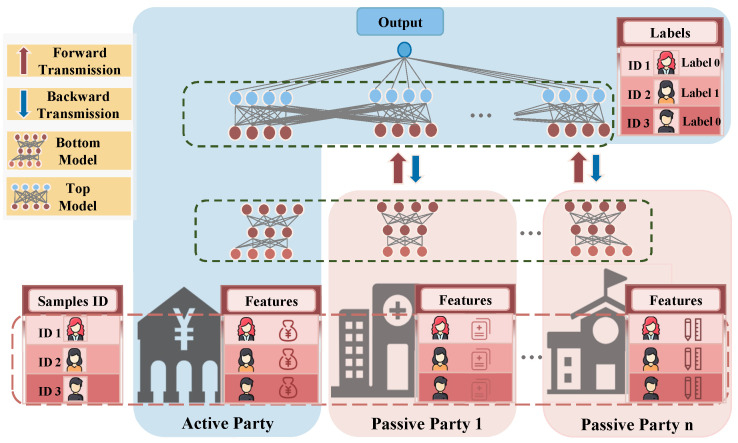
The Active Party owns the labels of the training data and deploys a top model, while the others are Passive Parties that only have features in local datasets and each of them deploys a bottom model. Each batch of VFL training requires two communications: a Passive Party passes its embedding to the Active Party, who then sends back its gradient to the Passive Party.

**Figure 2 entropy-27-00066-f002:**
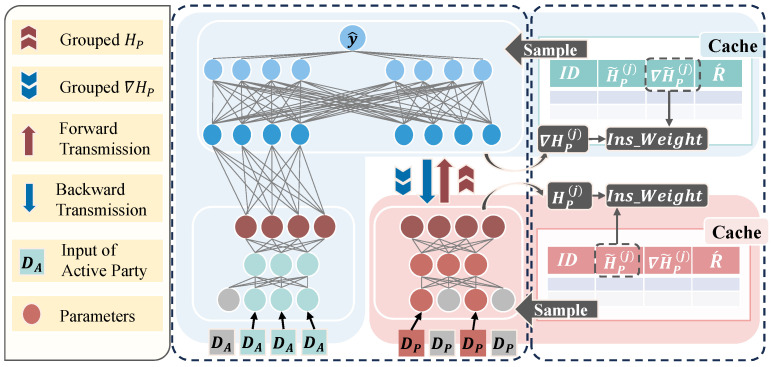
The framework of VFL-Cafe. Inputs or parameters in gray indicate features removed via selection. Each round, parties exchange *N* intermediate results, storing them in their own caches. When a party performs a local update, e.g., the Passive Party, it samples an instance from its cache, calculates the cosine similarity between the current and cached representations (${\tilde{H}}_{P}$ and $H_{P}$), and uses the weighted gradients ($\nabla {\tilde{H}}_{P}$) from the cache to update. $\overset{´}{R}$ denotes the times for local updates.

**Figure 3 entropy-27-00066-f003:**
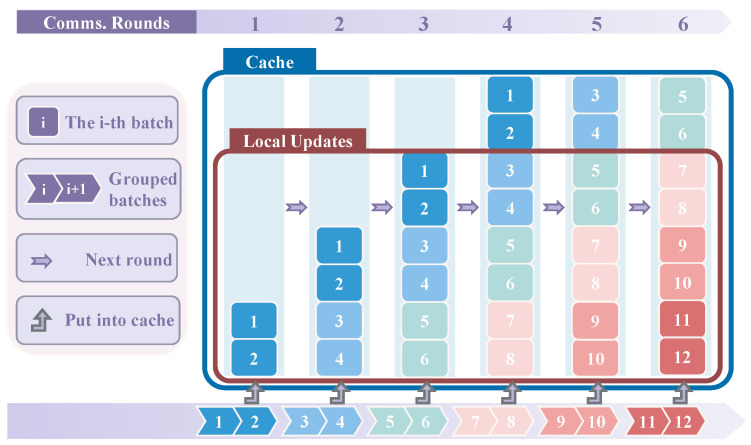
Set cache with parameters W=8, R=3, Q=6, and N=2. Each blue column represents the same cache across different communication rounds. After each round, the intermediate results (triangle boxes) of *N* mini-batches (with different colors indicating results from various rounds) are stored in the cache. The brown box labeled “Local Updates” indicates the total number of updates (*Q*) ≤6, allowing a maximum of six triangle boxes to be selected within it.

**Figure 4 entropy-27-00066-f004:**
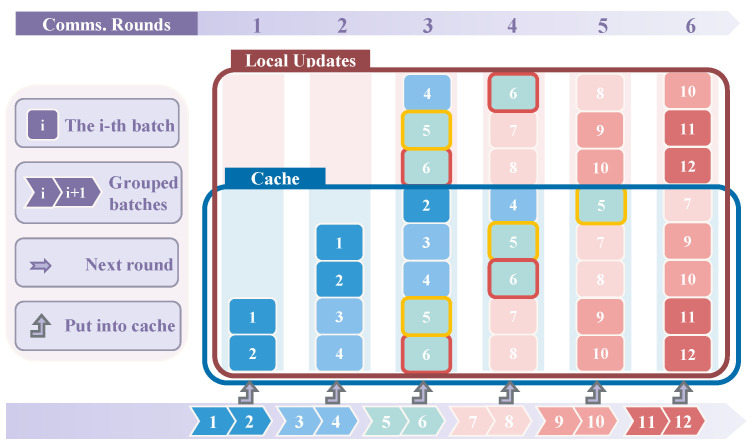
Set cache parameters W=5, R=4, Q=8, and N=2. Each blue column represents the cache (main window) across communication rounds, while each pink column denotes the virtual window, where local updates use batches from the current round’s main window. After each round, the intermediate results of *N* mini-batches are stored. The brown box labeled “Local Updates” indicates the total updates (*Q*) ≤8, allowing a maximum of five triangle boxes.

**Figure 5 entropy-27-00066-f005:**
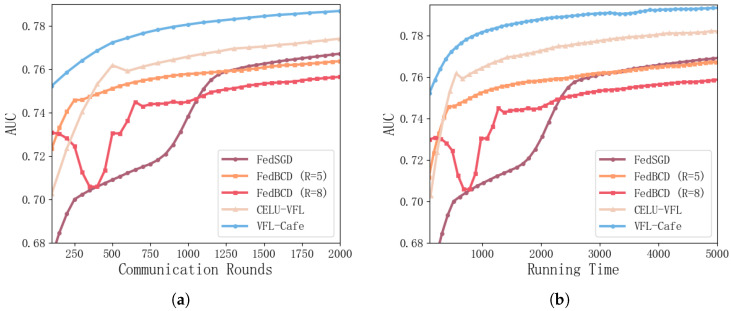
AUC curves over communication rounds and running time for our method compared to FedBCD (*R* = 5), FedBCD (*R* = 8), CELU, and FedSGD. (**a**) Validation AUC metrics in terms of communication rounds. (**b**) Validation AUC metrics in terms of running time (in seconds).

**Figure 6 entropy-27-00066-f006:**
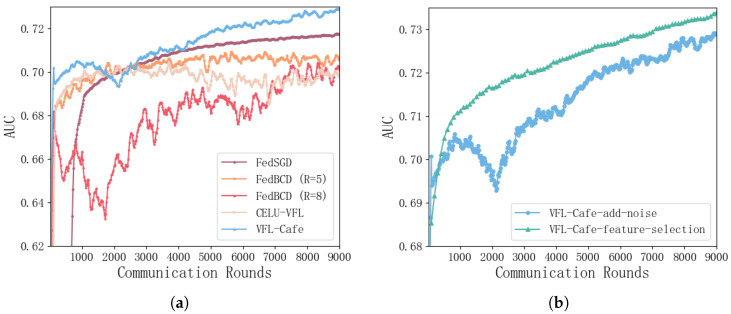
Comparison of AUC values before and after feature selection with *VFL-Cafe*, along with validation AUC metrics across communication rounds for our method compared to FedBCD (*R* = 5), FedBCD (*R* = 8), CELU, and FedSGD under added noise conditions. (**a**) Validation AUC metrics across communication rounds under added noise conditions. (**b**) Comparison of AUC metrics before and after feature selection with *VFL-Cafe*.

**Table 1 entropy-27-00066-t001:** Effect of SBE supplementation on body weight, body composition, and water and diet intake.

**Technique**	**Parameters**	***N* = 1**	***N* = 2**	***N* = 3**	***N* = 4**
Transmission of Multiple Mini-batches	W=8,R=5	14,956 ± 369.4	14,509 ± 357.3 (↓2.99%)	8095 ± 278.6 (↓45.80%)	7957 ± 229.5 (↓46.87%)
**Technique**	**Parameters**	R=1	R=3	R=5	R=8
Local Updates	N=1,W=5	35,076 ± 813.1	14,393 ± 425.4 (↓58.97%)	14,156 ± 387.0 (↓59.64%)	13,995 ± 399.5 (↓60.10%)
**Technique**	**Parameters**	W=1	W=3	W=5	W=8
Round-Robin Local Sampling	N=1,R=5	26,931 ± 636.2	15,483 ± 298.2 (↓42.51%)	16,950 ± 484.7 (↓44.47%)	14,956 ± 369.4 (↓37.06%)

When applying the cosine similarity mechanism for weighting, we consistently use an angle threshold $\zeta =60^{\circ }$ for comparison.

## Data Availability

The datasets used in this study are publicly available at https://www.kaggle.com/datasets/mrkmakr/criteo-dataset (accessed on 12 January 2024).

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
