# Peer review of "VFL-Cafe: Communication-Efficient Vertical Federated Learning via Dynamic Caching and Feature Selection"

_entropy, 2025, doi:10.3390/e27010066_

Round 1
Reviewer 1 Report
Comments and Suggestions for Authors
This paper proposed a vertical federated learning method namely VFL-Cafe that use dynamic caching and feature selection to boost communication efficiency and accuracy. More specifically, the caching scheme allows multiple batches of intermediate results to be cached and strategically reused by different parties, reducing the communication overhead while maintaining model accuracy even in Gaussian noisy environment. However, some issues are noticed by the reviewer as follows:
1. Why did author only choose the dataset public Criteo dataset for advertising click-through rate prediction?
2. In the proposed VFL framework, the detailed neural network architecture is not given. They just mentioned DLRM and WDL. The details of the chosen loss function are not given. Thus, it is not clear how did the model achieve accuracy?
3. To introduce noise why did author choose only Gaussian noise? Additionally, mean and variance value in the experiment is not clearly mentioned? Thus, it is not clear how did the model behaved with large uncertainty bound.
4. How did proposed caching system is secure?
5. Acronyms are well defined. Some defined acronyms like DLRM is not used further.
6. Author should clearly mention the internal neural network architecture for top and bottom model with activation functions.
Reviewer 2 Report
Comments and Suggestions for Authors
The authors of this manuscript introduce a framework that targets the reduction of the communication overhead imposed in vertical federated learning deployments by examining the introduction of local caching and noise wrangling.
The motivation backing the manuscript is strong and the work attacks a real-world problem. The scope is also relevant to the journal. moreover, the introduction and the problem formulation are easy for the average reader to grasp upon.
Some comments that must be addressed:
- the authors introduce the algorithmic methodology of their work but do not actually provide any implementation details for the framework and do not outline how the framework is actually used by potential users (configurations, services, api, etc). this is a clear shortcoming when space is clearly not an issue.
- the authors present the experimentation which targets FL but from my understanding only 2 clients are actually used. Why? A FL experiment should clearly feature a greater number otherwise what is the practical nature of the experiment to prove technical soundness?
- there is no overhead study conducted to show the computational overhead imposed to ease local client when embracing the framework in contrast to simply using a default VFL testing scenario.
- Are experiments actually run in a heterogeneous environment? why not use a testing suite like FedBed to evaluate the work?
Round 2
Reviewer 1 Report
Comments and Suggestions for Authors
Thank you for addressing all of my comments. I have no further comment.
Author Response
Thank you very much for your kind response and for taking the time to review the revised version of our manuscript. We are grateful for your positive feedback, and we appreciate your acknowledgment of the revisions made. It is truly encouraging to know that the changes we made address your concerns.
Thank you once again for your valuable time and constructive input.
Reviewer 2 Report
Comments and Suggestions for Authors
The manuscript in its current form constitutes a clear improvement over the initially submitted version and all comments of concern have been addressed.
I only have one comment. While the authors document that the dataset used is publicly available, the authors have not provided a link to a working prototype nor a code repository so that the framework can be used by researchers and even reproduce and verify the experiment results.
Author Response
Thank you very much for your constructive feedback and for recognizing the improvements made in the revised manuscript. We are grateful for your careful consideration of our work.
In response to your suggestion about providing access to the framework, we have now included the link to the code repository in the "Experiment Setup" section of the manuscript. We are currently in the process of finalizing and organizing the code, and once it is fully ready, we will upload it to the repository. We aim to make the code publicly available as soon as possible.
Thank you again for your valuable input.